# A Forward Future-Based Approach to Optimizing Agriculture and Climate Change Adaptation in Lower Eastern Kenya

**Lilian Wangui Ndungu** [1,*], **John Bosco Kyalo Kiema** [1], **David Nyangau Siriba** [1], **Denis Macharia Muthike** [2] and **Samuel Wamathai Ndungu** [3]

1   Department of Geospatial and Space Technologies, University of Nairobi, Nairobi P.O. Box 30197, Kenya
2   Mortenson Center in Global Engineering, University of Colorado, Boulder, CO 80309, USA
3   School of Pure and Applied Sciences, Kenyatta University, Nairobi P.O. Box 43844-00100, Kenya
*   Correspondence: lw_ndungu@students.uonbi.ac.ke; Tel.: +254-714-447-273

**Abstract:** Kenya's vulnerability to climate variability and change has been compounded by dependence on rain-fed agriculture with constrained capacity to adapt, a rapidly growing population, low-mechanized and low-input smallholder agricultural systems, and compromised soil fertility. The Ukraine war, COVID-19 and the desert locust invasion have only amplified the prevailing sensitivity to shocks in the agriculture sector, creating an emphasis on the need to strengthen local agricultural production to reduce reliance on imports. This paper seeks to assess the opportunities for improving agriculture adaptation and resilience based on future expected changes in climate, length of the growing period and agro-ecologies. The study uses 2020 as the baseline year and explores changes in agro-ecological zones (AEZs) in "near future" 2040 through two representative concentration pathways, 4.5 and 8.5, representing a medium carbon emissions and a dire emissions future, respectively. Google Earth Engine and R Statistics are used in data-processing. Down-scaled climate projections from CIMP5 are used for future analyses combined with static soil suitability and drainage data. Fuzzy logic is used to normalize inputs and compute the agro-ecological zones (AEZ). Interesting results emerge from the study that validate the hypothesis that the seasons and production potential are shifting. Lowland drylands will experience an increasingly long growing period, creating the potential for diversifying production systems from rangelands to agro-pastoral systems, with the capacity to grow more drought-resistant crops and the potential to take advantage of increased runoff for water harvesting. Midland highland areas, which form part of the food basket areas, have already started experiencing a reduction in the length of the growing period and agricultural potential. In these areas, resilience mechanisms will need to consider the expected future reduction in rain-fed agricultural potential, gendered preferences, convergence of technology and indigenous coping mechanisms, and drought-resilience-focused diversification.

**Keywords:** AEZ; Kenya; agro-ecologies; lower eastern; adaptation; resilience; predictions; future; climate change



## 1. Introduction

### 1.1. Background

There is no denying that the climate is changing, and fragmented areas with strained economies and coping mechanisms are most affected. The rising frequency and intensity of precipitation variations, heat waves, droughts and floods have all served as catalysts to rethink and re-evaluate climate adaptation investments, decisions and priorities [1–3]. The Inter-Governmental Panel on Climate Change (IPCC) has reported the alarming increase in greenhouse gas (GHG) emissions over the years, with agricultural production and its effects on land use contributing to these detrimental increases [4,5]. The impacts of climate change are expected to worsen in environments where other stressors, such as inflation, poverty and land size, amplify food insecurity [6,7]. The demand for food

and resource consumption will continue to create a destructive loop where increased competition for dwindling resources will trigger more degradation and increase losses in livestock, livelihoods and human lives [8]. In Kenya, agriculture is the backbone of the economy, contributing to almost 33% of gross domestic product (GDP) and employing more than 40% of the workforce that represents 70% of the rural population [9]. The evidence is overwhelming that the shorter recovery period between extreme events will worsen poverty, trapping smallholder farmers in a cycle of vulnerability [2].

### 1.2. The Complex Case of Food Insecurity

Already, the capacity for sufficient food production has been affected by cyclic droughts, floods and unexpected disruptions to regular weather patterns that have resulted in high human and livestock mortalities, as well as reduced productivity [10–12].

Over 3.1 million Kenyans faced acute food shortages in 2022, following a third consecutive below-average rainy season that led to deteriorating food security outcomes driven by the impacts of poor crop and livestock production, resource-based conflict, livestock disease and mortality, and the COVID-19 pandemic [13]. The sector's capacity to address food security has also been weakened by the disruption of agricultural supply chains and markets caused by the global financial and economic crisis [1,14].

The Ukraine war occurred when the country was already grappling with the effects of the COVID-19 pandemic and the desert locust invasion exacerbated the situation further, resulting in record high food prices due to shortages, and pushing millions more people into severe poverty and hunger. Years of development progress were undone, with food prices reaching all-time highs due to conflict in Ukraine, supply-chain disruptions, and the ongoing economic effects of the COVID-19 pandemic [1].

The changes in climate are expected to continue into the future with implications for agricultural production. By 2030, it is expected that temperatures will rise by 1 to 2.5 degrees Celsius. By 2050, there is consensus across the prediction models that the most evident change will be an increase in the average land surface temperature, estimated at between 3 to 7 °C, resulting in drier conditions and, in some cases, overriding the expected benefits that could be derived from the projected increase in precipitation in some areas [5,15]. By the end of the 21st century, climate change is predicted to raise the average temperature by 1.4 to 5.5 °C and the average amount of precipitation by 2 to 20 percent.

Projections also point towards slightly greater warming in the long-rains than in the short-rains season [5,16].

These changes are expected to result in average declines of around 12 percent in the output of rain-fed crops, including wheat, maize and rice in Africa by 2080, with a potential loss of 47% of agriculture revenue to intra-extra production and trade dynamics globally. In Kenya rain-fed agriculture yields could be cut by half and crop net sales could decrease by 90% by 2100 [4]. Even in highland areas, increased heat is predicted to reduce crop yields, leading to increased levels of food insecurity, increasing risks in the agricultural sector, the majority of which will occur through human-induced factors [4].

While numerous efforts have been made to increase Kenyan communities' ability to adapt to climate change, commensurate success for investments in climate adaptation and resilience has not been realized. If the current trajectory continues, smallholder farmers will suffer greater impacts from emerging climate-change-related problems [17].

### 1.3. Technology Driven Data for Decision Making

The increasing availability of satellites and Earth observation (EO) datasets and systems provides opportunities for evaluating past, present and future trends at micro-scales. High-quality, high-resolution remote-sensing and model-based datasets that are blended and unbiased that use station and ground data are now readily available and provide opportunities to improve the quality of information available for decision making [15]. This paper, seeks to evaluate what these climate shifts represent in terms of agricultural

potential, using predicted agro-ecological zones and the length of the growing period, and to identify opportunities for future-based agriculture and climate adaptation.

### 1.4. Global Climate Models

The primary tools for predicting climate change are global climate models (GCMs), which are developed based on consideration of alternative scenarios for the evolution of GHGs and aerosol concentrations. They work well at replicating both global and continental climate characteristics, such as global and continental temperature and precipitation patterns, and are intended to assess the behavior of the global climate system [18]. The most recent GCMs from the phase 5 Coupled Model Intercomparing Project (CMIP5) used the UK Meteorology Office Hadley Centre Global Earth System Models (HadGEM-ESM) and contributed to the Fifth Assessment Report of the Inter-Governmental Panel on Climate Change (IPCC AR5) [19]. Transparent assumptions about how the GCM/ESM has been derived are essential for diagnosing the simulated climate response and comparing responses across different models, especially since the implementation can involve subjective choices and may vary between modeling groups carrying out the same experiment [19]. This necessitates the need to downscale projections to improve their accuracy in informing scale specific decisions.

### 1.5. Defining Future Emission-Based Trajectories

CIMP5 contains outputs for the three different representative concentration pathways (RCP), defined as 2.6, 4.5 and 8.5, respectively. RCP4.5 depicts a medium emissions scenario in which greenhouse gas levels gradually rise until around 2040 before declining later. This scenario presupposes that climate policy will be used to alter related reference situations. The direst scenario, known as RCP8.5, predicts a future with steadily rising greenhouse gas levels [12,15,19].

## 2. Mapping Future Agro-Ecological Zones

The Food and Agricultural Organization (FAO) has developed methodologies for mapping agro-ecological zones [20] that were used to develop the Kenya Farm Management Handbook. To create comparable decision support outputs, the FAO methodology was adopted and adjusted to use satellite data, cloud computing and future climate scenarios, as shown in Figure 1. The year 2020 was used as the baseline year with AEZ computed using predicted climate parameters and current biophysical parameters for 2040 under RCP 4.5 and 8.5 [21]. Predicted climate regimes were combined with the static biophysical land inventory to produce the predicted AEZ. Fuzzy logic was used to compute the AEZ and to normalize inputs into the climate regimes and land inventory. Climatic and land constraints to agriculture production were included in the form of an aridity index and soil/slope constraints to agricultural production.

### Data Processing

Downscaled CIMP5 data for rainfall and temperature for 2040 were acquired and used due to their notably better performance, especially over eastern Kenya [15]. RCP8.5 and RCP4.5 were used in this study, since the two scenarios generally capture the potential range of future greenhouse gas releases and because fewer simulations are available for other emission scenarios [22]. The CIMP5 rainfall and temperature was used to derive potential evapotranspiration (PET), using the Thornthwaite (1948) equation, moisture regimes and the aridity index. Kenya's rainfall is bimodal, with two distinct wet seasons that support most of the rain-fed agriculture production [15,23]. This is defined by the Kenya Meteorological Department (KMD),as the long rains (LR) experienced through March, April and May (MAM) and the short rains (SR) occurring through October, November and December (OND).

The length of the growing period (LGP) was defined and computed as the number of days when

$$Tmean \geq 5^0C \tag{1}$$

and precipitation is

$$ppt > 0.5pet \tag{2}$$

The moisture regime is useful in capturing the seasonal variation in effective moisture and was computed using the Thornthwaite and Mather (1955) method based on annual rainfall and potential evapotranspiration [24]. The aridity index from Terra Climate was used as an indicator of the moisture deficit constraints to rain-fed production [25].

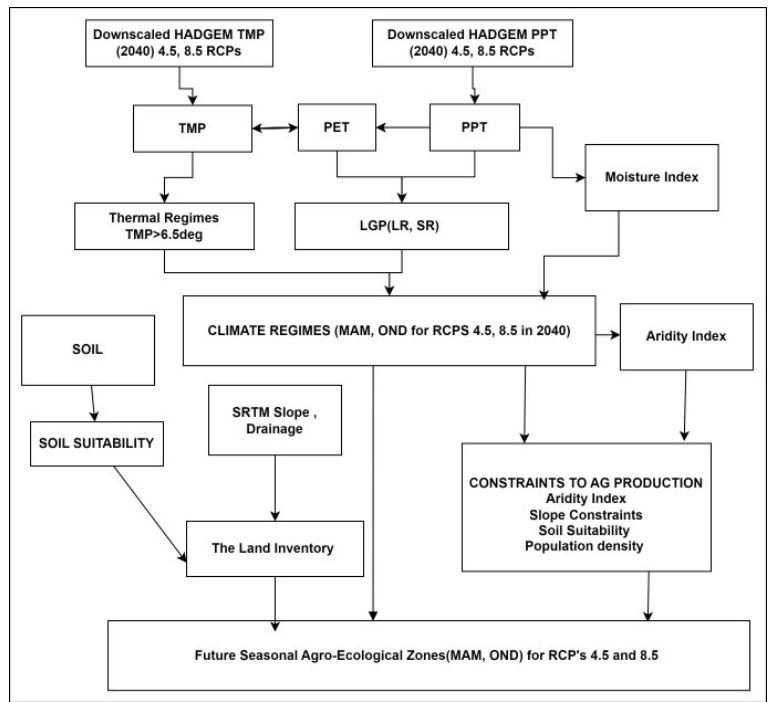

**Figure 1.** Methodology: An illustration of the approach used for mapping future agro-ecological zones.

Baseline period biophysical parameters, including soil suitability and drainage, were used and assumed to remain constant. Soil suitability for agricultural production was categorized from the Kenya Soil Survey (KSS) map and used to define soil characteristics, including texture, and their relationship to supporting agricultural production. The soil type, drainage capacity, elevation, depth and consistency were used in defining suitability for agricultural production [20,26]. The drainage map was derived from Shuttle Radar Topography Mission (SRTM) elevation data [27].

## 3. Results

### 3.1. Changing Agro-Ecological Potential in a Changing Climate

To capture micro-variations, all layers were acquired at a scale of 1 km or less where possible. Fuzzy logic was used to normalize the outputs. Change was determined from the 2020 baseline assessments and the 2040 predictions by season (MAM and OND). From the assessment, the expected climate shifts and increase in precipitation was observed, with higher gains observed during OND than MAM. This corresponds to RCMRD's [15] assessment that future seasonal rainfall will likely increase during OND under all scenarios, with greater increase over eastern parts of Kenya, as shown in Figure 2.

Temperature was adjusted to thermal regimes by adding a threshold to allow for adjustment of temperatures that fall below the minimum needed for crop growth [26]. Increasing temperature was observed across the region, with higher increases observed in the lower eastern part, but more pronounced increases in MAM than in OND, as seen

in Figure 3. This is consistent with the consensus on projections that suggests changes in temperature point to a warmer future over almost all parts of Kenya. In the near future, the annual surface temperature is projected to increase by between 1.0 °C and 2.0 °C under the RCP4.5 scenario, but will likely be greater in the RCP8.5 scenario, which is expected to be between 1.5 °C and 2.5 °C [15]. Most of the change in thermal regimes was observed from the lowland regions in Machakos and in the highland areas of Tharaka Nithi and Meru, with the change pronounced in both MAM and OND.

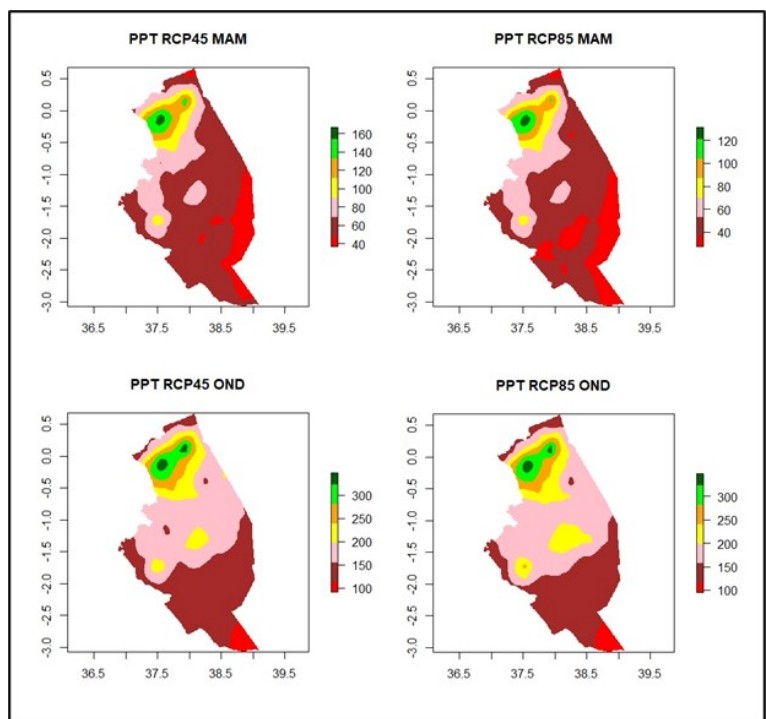

**Figure 2.** Precipitation range across MAM and OND for 2040 RCP's 4.5 and 8.5.

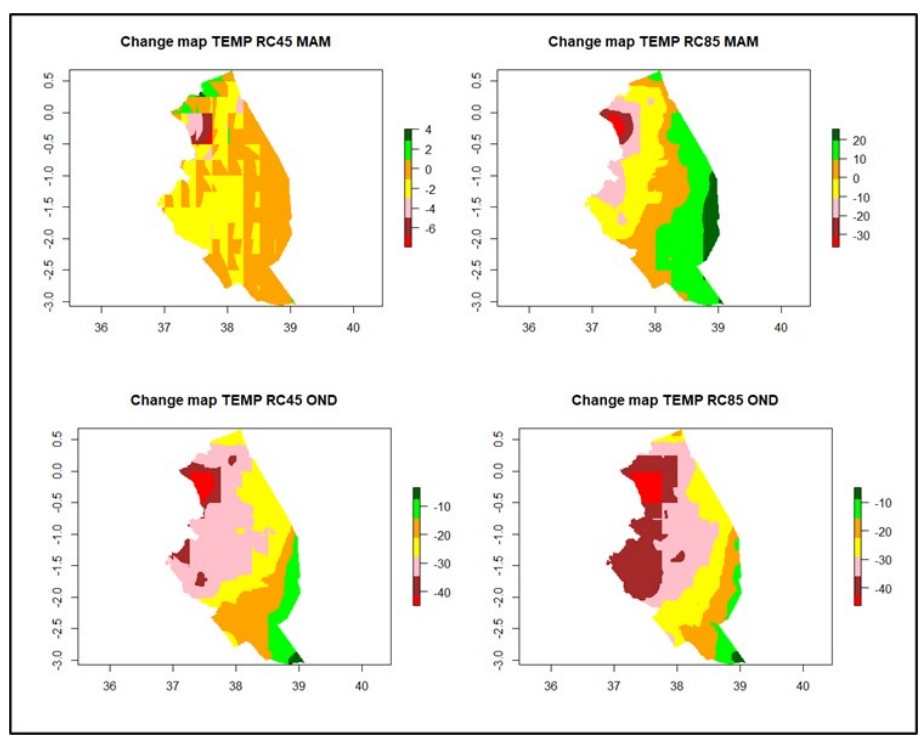

**Figure 3.** Thermal Regimes for MAM, OND in 2040 for RCPs 4.5 and 8.5.

An increase in PET adversely affects agriculture, as more water is lost to evaporation and transpiration. An increase in PET was observed across the study area, with a more pronounced increase in RCP 8.5 than in 4.5, as seen in Figure 4. The highest increases were observed in already constrained areas, such as Embu, Tharaka Nithi, North Meru and the lower sides of Kitui.

Increases of between 1–20 growing days were observed in Embu, Tharaka Nithi, Embu, Kitui, Machakos and Makueni in the MAM period for RCPs 4.5 and 8.5, with the lower drylands anticipated to experience a much higher increase in both MAM and OND, as seen in Figure 5. Parts of the north arid areas of Meru also showed an increase in LGP. However, the main growing areas in Meru showed a decrease in the length of the growing period of 1–10 days in MAM for RCP 4.5 and 8.5. OND projections for RCP 4.5 and 8.5 exhibited negative trends for the main growing areas in Meru. A higher increase in LGP OND over the lower eastern areas of Kitui and Machakos, and in the upper parts of Machakos and Embu, was predicted, where a 1–20 day increase was observed for RCP 4.5. However, the highland areas experienced a decrease in the length of the growing period. Similar but more pronounced trends were observed for RCP 8.5.

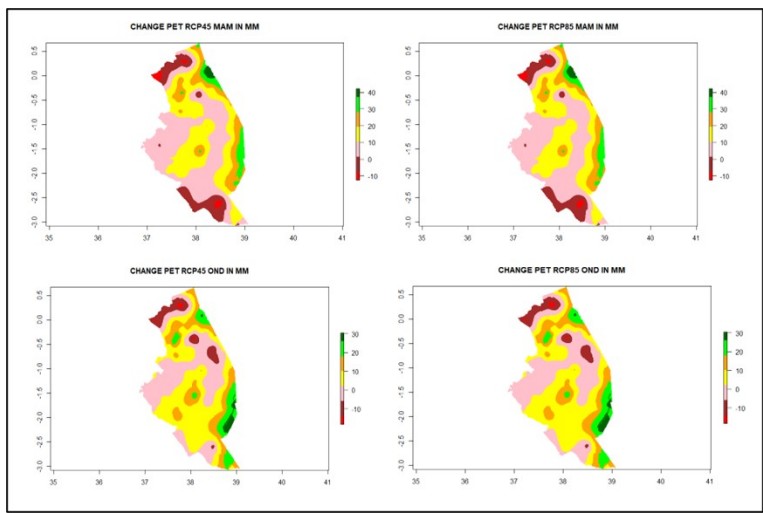

**Figure 4.** Potential evapotranspiration and change across the seasons and epochs.

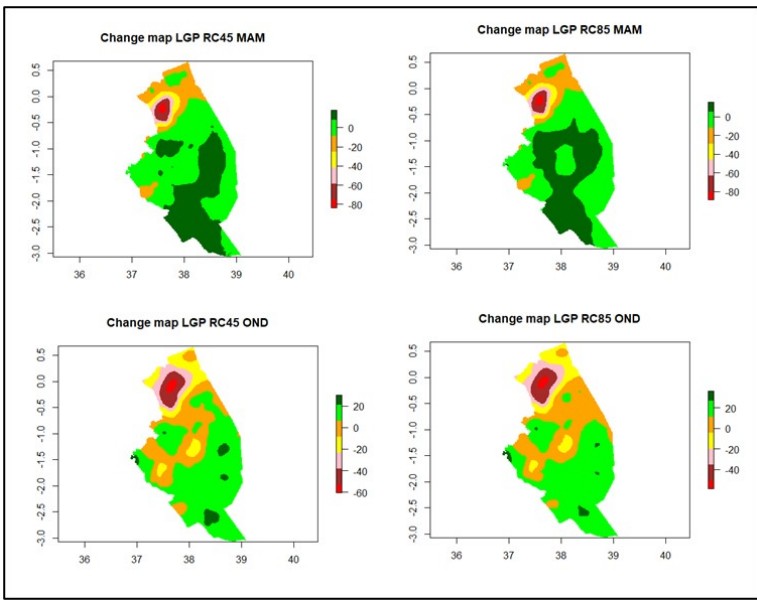

**Figure 5.** Changes in the length of growing days during MAM, OND in 2040 for RCPs 4.5 and 8.5.

From the aridity index, higher increases in effective moisture were observed in the eastern dryland regions, with the losses more pronounced in Machakos, Embu, Tharaka Nithi and Meru in OND, as seen in Figure 6 and 7 .

Climate regimes were computed from PET, with thermal regimes, moisture index and LGP computed by season. The inputs were first normalized using fuzzy logic before being combined to produce the climate regimes (Figure 8).

Soil suitability and soil drainage derived from the soil and elevation data indicate the biophysical suitability for agricultural production (Figure 9).

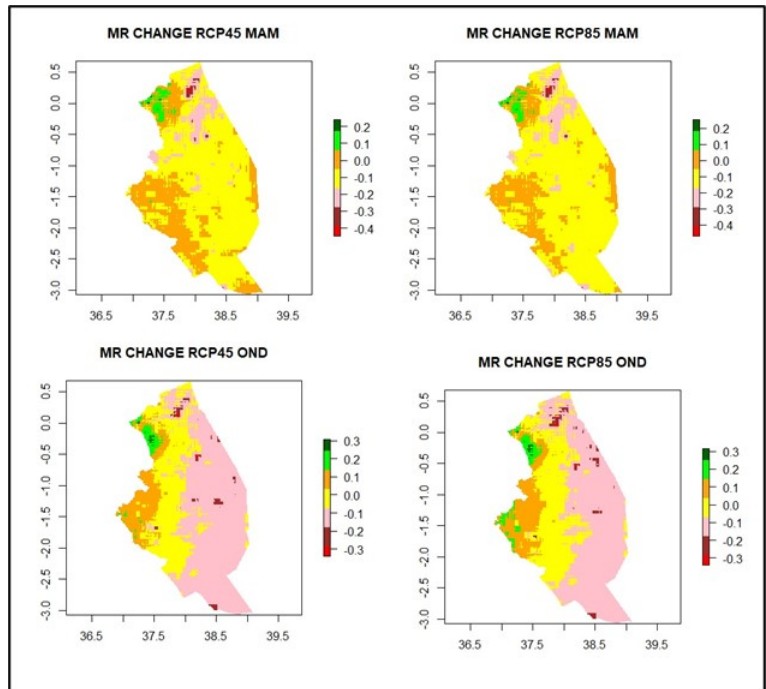

**Figure 6.** Changes in moisture index during MAM, OND in 2040 for RCPs 4.5 and 8.5.

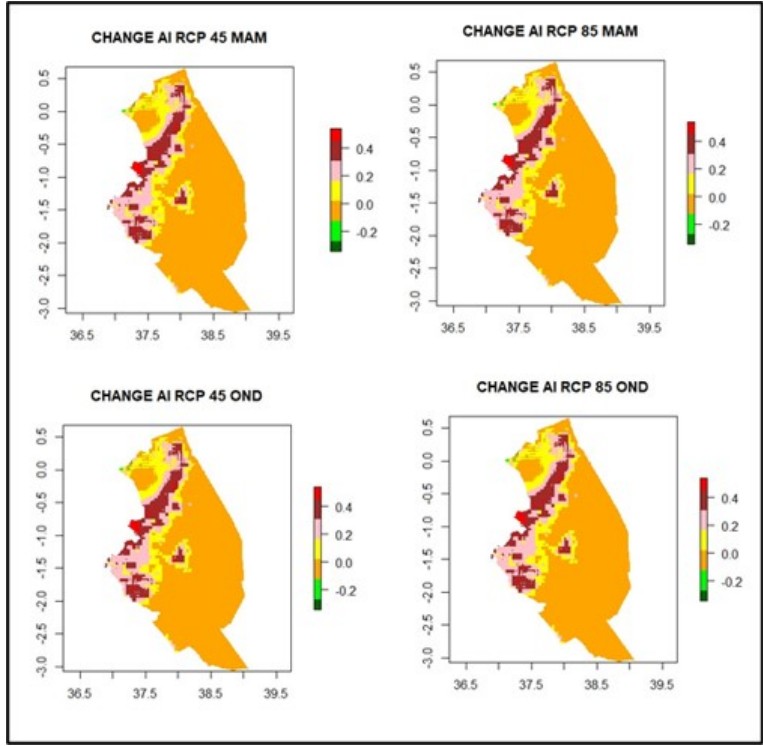

**Figure 7.** Changesin aridity index during MAM, OND in 2040 for RCPs 4.5 and 8.5.

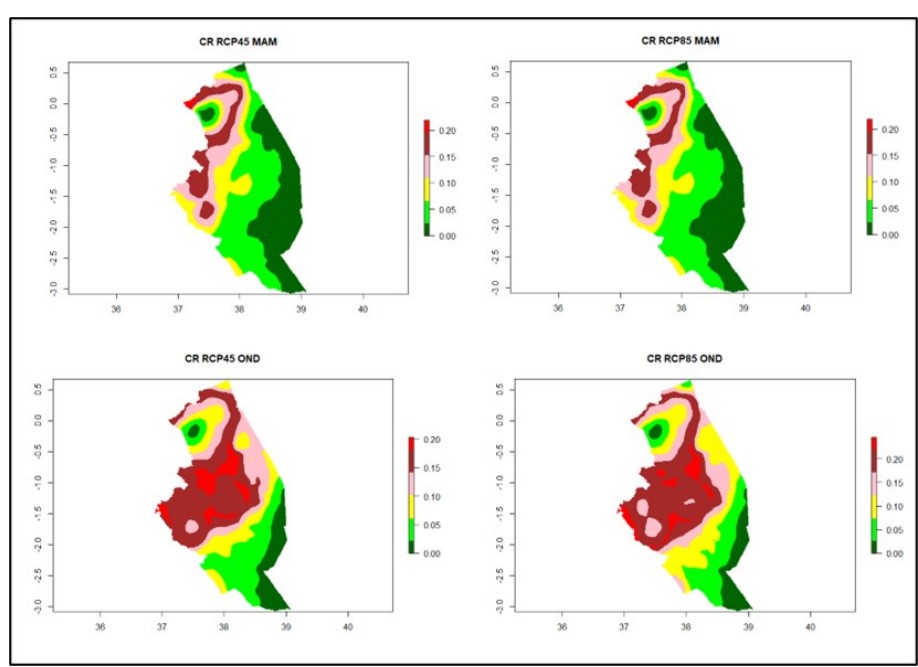

**Figure 8.** Climate regimes for MAM, OND in 2040 for RCPs 4.5 and 8.5.

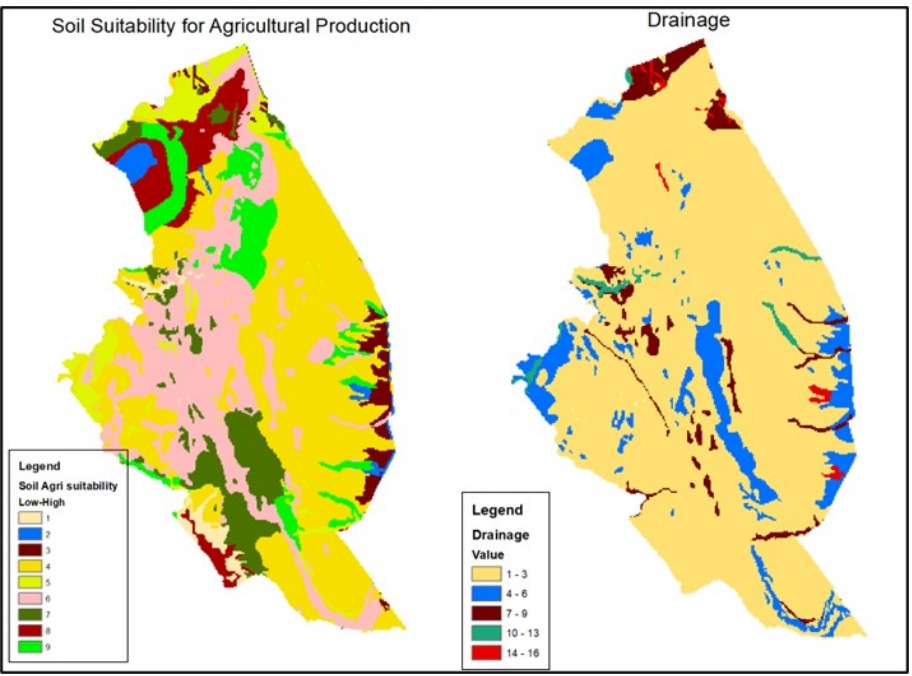

**Figure 9.** Soil suitability for agricultural production and drainage.

Climate regimes and biophysical parameters were normalized using fuzzy logic and used as inputs to the definition of the AEZ. Linear and inverse relationships were applied based on the variable's contribution to agriculture potential. An inverse relationship applies in the interpretation of AEZ change maps. For example, a change from 7 (per-arid) to 6 (arid) represents improved potential. The highest losses in potential of the agro-ecologies were observed around Machakos, Embu, Meru and Tharaka Nithi and were more pronounced in RCP 4.5 MAM than in RCP 8.5 MAM (see Figure 10). Improvements in AEZ were more notable during the OND for both RCPs in the lower eastern region, corresponding with the noted increase in rainfall and length of the growing period.

### 3.2. Learning from the Current Adaptation to Improve Forward Future-Based Investments

The baseline AEZ mapping covering 1990–2020 demonstrated a historical change, where an opposite shift was experienced, with highland and midland areas experiencing a reduction in LGP and AEZ potential, while drylands experienced an improvement. To verify this change, 860 farmers within the microclimates and agroecologies in the study area were interviewed.

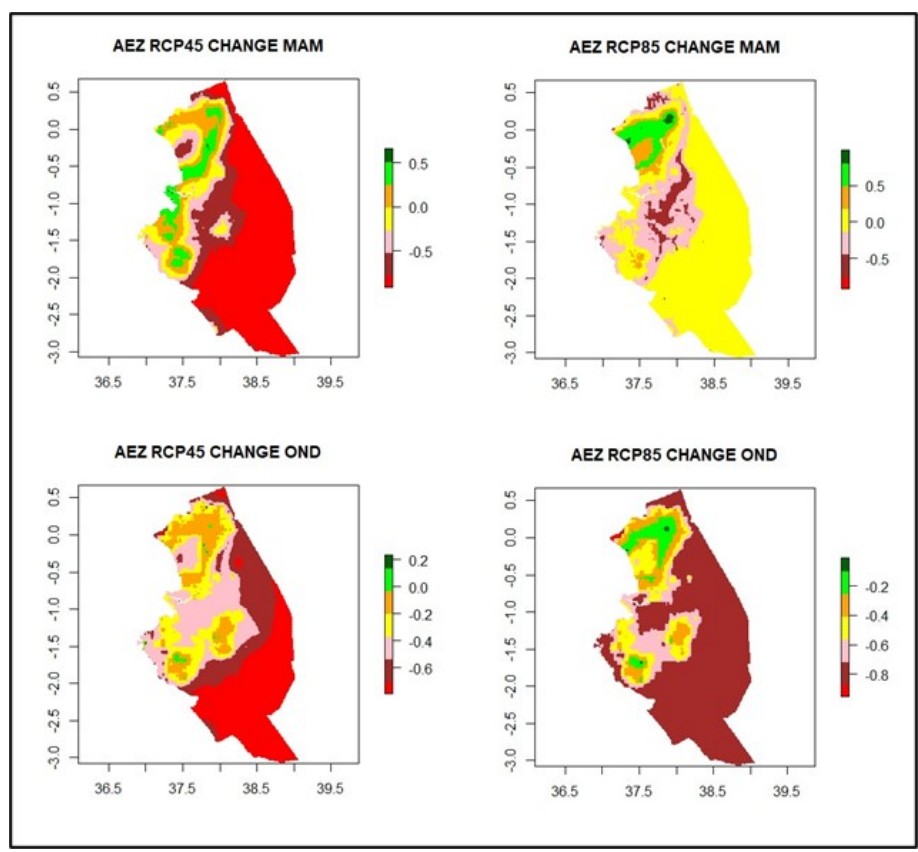

**Figure 10.** Agro-ecological zones and change maps from a 2020 baseline and RCP 4.5 2040.

Despite considerable investments by the government and private sector, as well as indigenous adaptation measures, food security was still a major concern in most of the households, despite diversification in farming systems and in spite of larger land holdings, as shown in Figure 11.

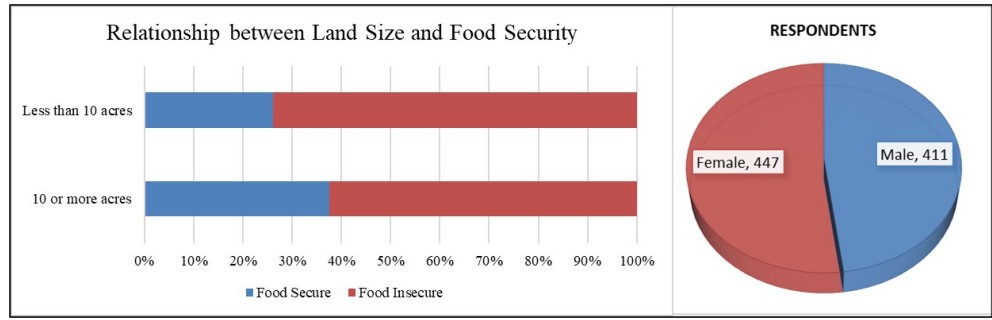

**Figure 11.** Relationship between land size and food security and demographics.

Different adaptation measures were being adopted, as seen in Figure 12. Most farmers adjusted their farming by adopting drought-resistant crops, adjusting planting dates and diversifying livelihoods to include livestock-keeping. Soil management focused on the use of manure, retaining residue and zero tillage. Water harvesting and conservation methods were also used to manage water availability.

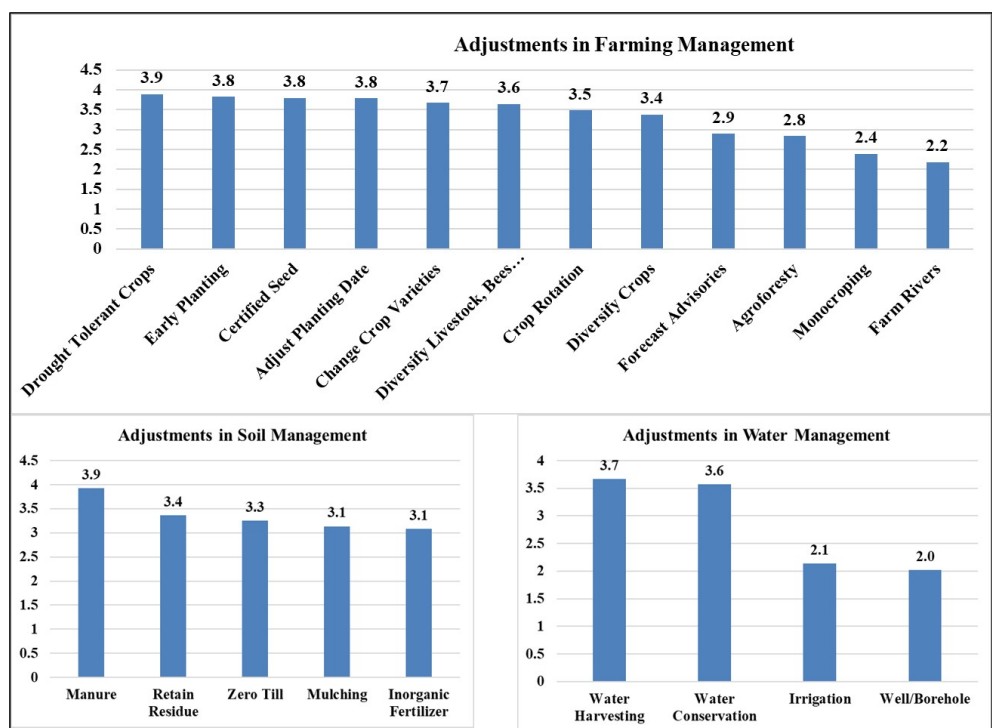

**Figure 12.** Farmers adaptation strategies for farming, soil and water management.

These adaptation mechanisms were enabled through farmers' initiatives as well as government and donor investments in the region. Lack of capital, lack of storage facilities and challenges accessing extension and education were identified as key barriers (Figure 13).

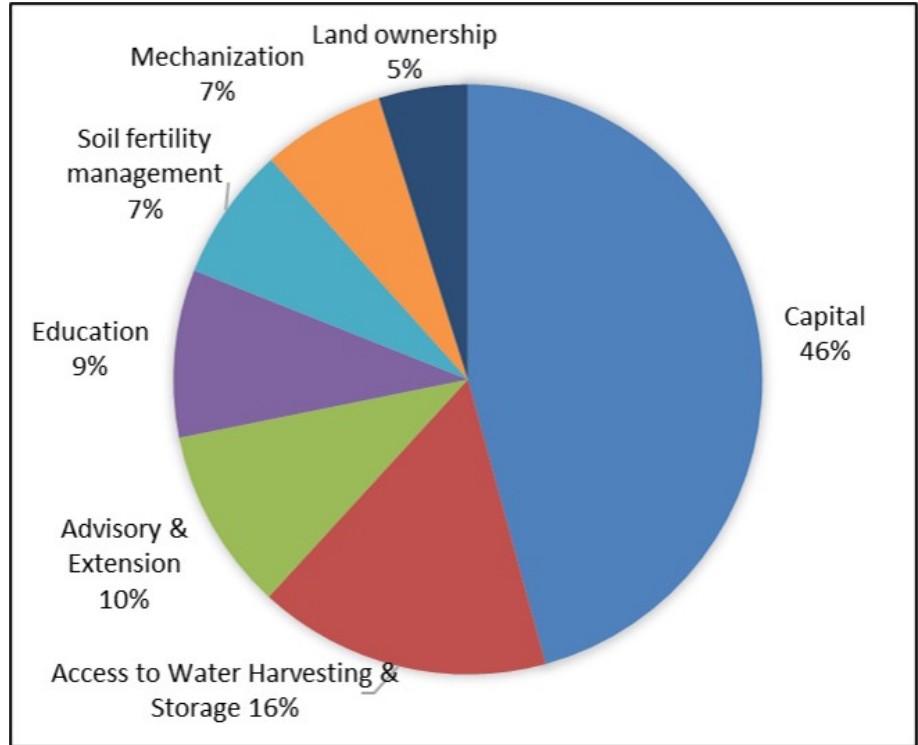

**Figure 13.** Farmers' perceived adaptation barriers.

### 3.3. Opportunities for Adapting and Transitioning Adaptation Strategies

### 3.3.1. Gendered Adaptation Mechanisms

Gender influences the success of adaptation mechanisms. For example, the survey found that female-headed households adapted by diversifying food crops, while male-headed households adapted by diversifying cash crops with reported higher incomes, as seen in Figure 14. Female-headed households were more interested in improving their food security first. Future adaptation mechanisms will need to be structured with the sensitivities of gender and population groups considered. For example, high-yielding, drought-resistant and early maturing food crops would be more appropriate for female-headed households.

| Analysis of Gender Vs. Income Sources Vs. Diversity of Crops | | | | |
|---|---|---|---|---|
| Respondent | Decision Maker | Sources of Income | Diversity of Food Crops | Diversity of Cash Crops |
| Female | Female | 2.47 | 3.62 | 1.73 |
| Female | Male | 2.16 | 3.20 | 1.30 |
| Male | Female | 2.44 | 3.40 | 1.70 |
| Male | Male | 2.48 | 3.17 | 1.94 |
| | *Median* | *2* | *3* | *1* |
| | *Minimum* | *1* | *0* | *0* |
| | *Maximum* | *5* | *7* | *5* |

**Figure 14.** Relationship between gender, adaptation bias and incomes.

### 3.3.2. Agricultural Land Consolidation and Zoning

From the survey, reported food insecurity in large areas (>5 acres) points to challenges in land utilization and optimization of farming activities to increase the return on investments. Agricultural zonation, where farmers collectively plant specific suitable crops for their agro-ecologies, has been used to optimize production and access to markets, improving the resilience of farmers. Further considerations for consolidation of land for agricultural production would allow leveraging of economies of scale to reduce production costs and increase return on investments in a sustainable agriculture business model. Moreover, this would allow responsive production to both local and international market demands.

### 3.3.3. Focused Diversification of Farming Systems

The projected increase in rainfall and the length of the growing period in the lowland drylands (from per arid/arid to arid/semi-arid) presents an opportunity to shift purely rangeland areas to agro-pastoral production. The study conducted by RCMRD [15] confirmed that higher run-off in OND was observed in lower dryland areas. This represents an opportunity for water harvesting for increasing agro-pastoral systems. In the midland areas of Embu, Tharaka Nithi and Meru, adaptation mechanisms would need to focus on the projected loss in agricultural potential to introduce production systems matching the reduced potential.

### 3.3.4. Customizing Adaptation Based on Vulnerabilities

An understanding of perceptions of drivers to vulnerability was found to be different between food-secure and food-insecure respondents. An assessment of the education levels of the respondents did not seem to affect their perceptions of their sensitivity to climate change. However, there were distinct variations in the definitions of the responses from food-secure and food-insecure households (Figure 15). For example, food-secure households responded that they were more affected by farm destruction by floods and post-harvest losses than drought, poverty or inflation. This implies that, in food-secure areas, management of post-harvest losses was more important as well as mitigation of flood-related destruction.

| Food Security | Secure | Insecure | Farmers perception on vulnerability drivers |
|---|---|---|---|
| Food crops Diversity | 3.19 | 3.13 | Food insecure persons have less food crops diversity by about 2% |
| Cash Crops Diversity | 1.58 | 1.41 | Food insecure persons have less Cash crops diversity by about 13% |
| No Positive Climate Change Outcome | 0.40 | 0.46 | Food insecure people find negative climate change to blame for their food insecurity |
| Impact on Production Trend | 0.97 | 0.95 | Food secure people are more impacted by climate change on their production. |
| Impact on Food Security | 3.22 | 3.66 | Food insecure people are more impacted by climate change |
| Season Change | 4.90 | 4.88 | Food secure people are more impacted by season change |
| Rain Time Change | 5.12 | 5.12 | Rain time change is insignificant to food security among respondents |
| Drought Crop Fail | 4.83 | 4.87 | Food insecure people are more susceptible to drought crop failure |
| Flood Farm Destruction | 3.61 | 2.28 | Food secure people were more affected by farm flood destruction |
| Post Harvest Loss | 4.56 | 3.69 | Food secure were more afffected by post harvest losses |
| Pests | 4.68 | 4.76 | Food insecure people were more affected by pests |
| Diseases | 4.50 | 3.59 | Food secure people were more affected by diseases |
| Poverty food shortage | 4.57 | 4.95 | Food insecure people are susceptible to poverty food shortages |
| Lack of Water | 4.78 | 4.29 | Food secure people identified lack of water as a significant climate change issue on their production |
| Soil Fertility Degradation | 4.82 | 3.46 | Food secure people identfied soil fertility degradation as an issue. *Probably food insecure people don't know how to assess soil degradation |
| Crop Viability | 4.66 | 3.74 | Food secure people identified climate change as a significant driver to reduced crop viability |
| Livestock Death | 4.11 | 2.81 | Food secure people are more lively diversified to livestock given the high impact they noted on livestock due to climate change |
| Inflation on Costs of Inputs | 5.53 | 5.72 | Food insecure people are highly vulnerable to inflation on cost of inputs |

**Figure 15.** Perceptions of drivers of food insecurity between food-secure and food-insecure households.

### 3.3.5. Converging Technology and Indigenous Knowledge for Adaptation

Most respondents accessed forecasts and used them to inform their farming activities. It is important to note that 27% of the farmers used indigenous knowledge of forecasts and season change to adapt their farming practices. The convergence of technological advances and indigenous knowledge can be harnessed to increase access to information for optimizing adaptation. Further, education did not emerge as a major barrier to access to forecasts since TV and radios provided vernacular translations, as seen in Figure 16.

### 3.3.6. Innovations for De-Risking Agriculture and Use of Technologies

There is an increasing investment in de-risking agriculture in Kenya, with the government-subsidized crop insurance extended to 33 counties, covering high, medium and marginal production areas. Lowland drylands have especially benefited from higher subsidies from the government. The top barrier was cited as lack of access to capital and extension/education, as well as mechanization.

Opportunities for innovative financing of farmers is required, as well as education, to ensure that there is a return on investment for the farmer. The increasing mobile and internet connectivity has made farmers more accessible. The African Union, through its digital agriculture strategy has emphasized the need for countries to register their farmers and develop agricultural data hubs that harness technological advances to deliver customized information, resources and opportunities for farmers [28].

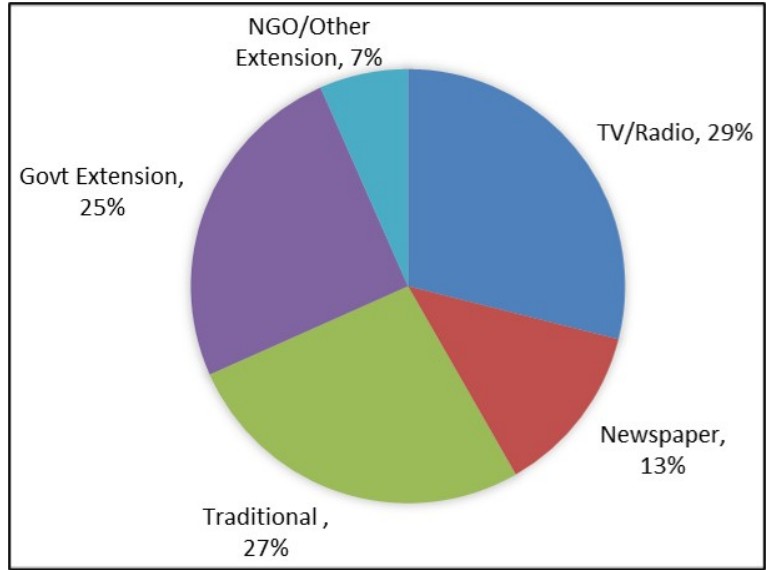

**Figure 16.** Access to information and forecasts irrespective of education level.

## 4. Conclusions

A changing climate presents new opportunities for optimizing adaptation mechanisms. This research has highlighted the expected changes in agricultural potential. The methods developed utilize data and approaches that are scalable for replication and learning in other areas. The understanding provided of the projected changes in agricultural potential can guide the prioritization of investments to effect the cultural change required to ensure that these investments result in increased resilience.

Further, the study has demonstrated the value of projection-based climate adaptation planning in providing opportunities for adjusting trajectories in investments for resilience building. While the study is not exhaustive, it focuses on demystifying the "gloom and doom" around climate change, focusing instead on the opportunities that climate change presents for adaptation. For sure, every cloud has a silver lining.

**Author Contributions:** Writing—original draft preparation, L.W.N.; writing—review and editing, L.W.N.; climate predictions data extraction, D.M.M.; statistical analyses, S.W.N.; supervision, review and editing, J.B.K.K. and D.N.S. All authors have read and agreed to the published version of the manuscript.

**Funding:** This research was possible with funding from the Kenya Climate Smart Agriculture Programme (KCSAP) https://www.kcsap.go.ke/ (accessed on 29 October 2022).

**Data Availability Statement:** All satellite data used is freely available online and through the Google Earth Engine.

**Acknowledgments:** I would like to acknowledge the support of Zacharia Mwai who I consulted with on the definition of the climate parameters. I would also like to thank Seth Nyawacha for the immense support provided and time invested to consult on automation of the workflows, especially in the use of GEE.

**Conflicts of Interest:** The authors declare no conflict of interest.

## Abbreviations

The following abbreviations are used in this manuscript:

| | |
|---|---|
| AEZ | Agro-ecological zones |
| AI | Aridity index |
| CIMP5 | Coupled Model Inter-comparison Project |
| EO | Earth observation |
| FAO | Food and Agricultural Organization |
| GCM | Global climate models |
| GDP | Gross domestic product |
| GEE | Google Earth Engine |
| GHG | Greenhouse gas |
| HadGEM-ESM | Hadley Centre Earth System Models |
| IPCC | Inter-Governmental Panel on Climate Change |
| IPCC AR5 | Fifth Assessment Report of the Inter-Governmental Panel on Climate Change |
| KMD | Kenya Meteorological Department |
| KSS | Kenya Soil Survey |
| LGP | Length of the growing period |
| LR | Long rains |
| MAM | March April May |
| MR | Moisture regimes |
| OND | October November December |
| PET | Potential evapotranspiration |
| RCP | Representative concentration pathways |
| SR | Short rains |
| SRTM | Shuttle Radar Topography Mission |

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
