# Peer review of "A Forward Future-Based Approach to Optimizing Agriculture and Climate Change Adaptation in Lower Eastern Kenya"

_land, doi:10.3390/land11122172_

Round 1
Reviewer 1 Report
This is a very interesting and highly pertinent paper.
While I am not qualified to independently judge the conclusions, the real strength in this paper is in any case in its approach.
1. What is the main question addressed by the research? The authors give an overview of their assessment of how climate change will affect agriculture in eastern Kenya, breaking it down by altitude and area to give projections and suggestions for mitigation and further research.2. Do you consider the topic original or relevant in the field? Does it address a specific gap in the field? The topic is obviously relevant, and while the research in itself in the paper is not groundbreaking, it is in my view more of a review paper, taking specialized technical information, summarizing it and putting it in context and making it available. Given the current situation with regards to food security and climate, it is very relevant and addresses a gap or need in public management and agricultural planning.
3. What does it add to the subject area compared with other published material? Not that many have focused on Kenyan agriculture using updated climate projections.
4. What specific improvements should the authors consider regarding the methodology? What further controls should be considered? You can always write more and do more specific models on yield from different land use and crop strategies. However, that is further research this one paper cannot hold everything without becoming a monograph or a book. As for controls, I am not sure how that question would make sense in this context.
5. Are the conclusions consistent with the evidence and arguments presented and do they address the main question posed? Yes. It is, and they do.
6. Are the references appropriate? As far as I can tell, but I admit to not having gone through them in any detail. (It didn't seem like a particular need.)
7. Please include any additional comments on the tables and figures. The text is generally very good, but a few unclear text passages that I can mention include: Page header: It says "Version October 31, 2022 submitted to Journal Not Specified" which seems odd, but I don't know if that is the authors or mdpi desk mistake. L60-66: Make clear if you are talking about Kenya or globally, and over the range of which scenarios L71: What is "intra-extra production"?
L90-L100: Mention why AR5, not AR6, is the basis for this paper.
L103: As a final touch bringing the paper up to date maybe mention the recent UNEP Emissions Gap Report 2022: The Closing Window ? L149 -151: The first part of this paragraph seems to belong in "Methods", not "Results". Move it?L150: Give a relevant reference or describe in a non-fuzzy way what your use of "Fuzzy logic" entails in this analysis L220: "The survey" is mentioned. Make clear that this is the KSS. (If it isn't the KSS you of course need to explain which survey you are referring to.) L276: This wins the award for most manically upbeat sentence of the year. Figures: Nice, but please include a larger-scale map where the area in question is highlighted so that people unfamiliar with Kenya may locate it. Tables: Please indicate the source of the data in the figures and tables in the captions.
Author Response
Review 1
Point 1.
The text is generally very good, but a few unclear text passages that I can mention include: Page header: It says "Version October 31, 2022 submitted to Journal Not Specified" which seems odd, but I don't know if that is the authors or mdpi desk mistake.
Response: Resolved
Point 2
L60-66: Make clear if you are talking about Kenya or globally, and over the range of which scenarios
Response- Projections also point towards slightly larger warming in the long rains than in short rains season in Kenya
Point 3
L71: What is "intra-extra production"?
Response Clarified in the text to mean internal and external production dynamics
Point 4
L90-L100: Mention why AR5, not AR6, is the basis for this paper.
Response Justification provided.
“Trade-offs in selection of CIMP5 was based on a comparison assessment that established that CIMP6 demonstrated lower performance in simulation spatial variability for most of the climate variables and timescales\cite{kamruz2021}. Further, CIMP5 down-scaled projections over Kenya were available for this localized study.”
Point 5
L103: As a final touch bringing the paper up to date maybe mention the recent UNEP Emissions Gap Report 2022: The Closing Window ?
Response
Recommendation added.
“But their value in changing the trajectory of climate change. The latest report from UNEP(United Nations Environment Programme), reveals that there has been inadequate progres on climate action, and that to achieve gains that get the world on a least-cost pathway to holding global warming to 1.5°C, emissions must fall by 45 per cent over those envisaged under current policies by 2030. For the 2°C target, a 30 per cent cut is needed. Such massive cuts mean that large-scale, rapid and systemic transformation across key sectors and systems such as agriculture, are needed\cite{logan2022}.”
Point 6
L149 -151: The first part of this paragraph seems to belong in "Methods", not "Results". Move it?
Response: Sentence moved to methods
Point 7
L150: Give a relevant reference or describe in a non-fuzzy way what your use of "Fuzzy logic"
Response
Section added describing the value in the linking fuzzy and non fuzzy outputs.
\subsection{Opportunities for adapting and transitioning adaptation strategies}
To understand the trajectory, current adaptation mechanisms were evaluated against the future projections to identify opportunities for adapting and transitioning adaptation strategies. With the results from the predictive mapping providing an expectation of the trajectory in future changes in climatic and agro-ecological potential, this section provides an analysis that builds on these predictions and ground surveys to recommend opportunities for optimizing forward future based adaptation.
Point 8
entails in this analysis L220: "The survey" is mentioned. Make clear that this is the KSS. (If it isn't the KSS you of course need to explain which survey you are referring to.)
Response
A section describing the survey was added Line 164-185
Point 9
L276: This wins the award for most manically upbeat sentence of the year. Figures: Nice, but please include a larger-scale map where the area in question is highlighted so that people unfamiliar with Kenya may locate it.
Response
Thank you (takes a bow). A section on the study area with the requested map added.
Point 10
Tables: Please indicate the source of the data in the figures and tables in the captions.
Response
Figures where data is not derived have the description added. Precipitation and temperature downscaled predictions were used to derive the rest of the outputs from the methods outlined.

Reviewer 2 Report
Hello, dear author or/s
Thank you for your efforts on this manuscript.
The results of the work can be attractive to the audience.
I hope better software and programming will be used in your future work.
These climate data download sites do not have a high scientific citation and usually have less than 60% accuracy.
Best Regards

Author Response
Recommendations noted. Thank you.

Round 2
Reviewer 2 Report
Hello, dear Ndungu et al.,
The text of the manuscript has been greatly improved
I am very happy that this manuscript will be published in the next issue of the journal
Congratulations to my friends and colleagues at the University of Nairobi
